# Urban Therapy—Urban Health Path as an Innovative Urban Function to Strengthen the Psycho-Physical Condition of the Elderly

**DOI:** 10.3390/ijerph20126081

**Published:** 2023-06-08

**Authors:** Anna Szewczenko, Ewa Lach, Natalia Bursiewicz, Iwona Chuchnowska, Sylwia Widzisz-Pronobis, Marta Sanigórska, Klaudia Elsner, Daria Bal, Mateusz Sutor, Jakub Włodarz, Józef Ober

**Affiliations:** 1Faculty of Architecture, Silesian University of Technology, Akademicka 7, 44-100 Gliwice, Poland; sylwia.widzisz-pronobis@polsl.pl (S.W.-P.);; 2Faculty of Automatic Control, Electronics and Computer Science, Silesian University of Technology, 44-100 Gliwice, Poland; ewa.lach@polsl.pl (E.L.);; 3Institute of History and Archival Studies, Pedagogical University of KEN, Podchorążych 2, 30-084 Cracow, Poland; natalia.bursiewicz@up.krakow.pl; 4Faculty of Biomedical Engineering, Silesian University of Technology, Roosvelta 40, 41-800 Zabrze, Poland; iwona.chuchnowska@polsl.pl; 5Department of Applied Social Sciences, Faculty of Organization and Management, Silesian University of Technology, Roosevelta 26-8, 41-800 Zabrze, Poland; jozef.ober@polsl.pl

**Keywords:** active urban space, active ageing, senior citizens, physical activity, physical health, mental health, mobile application, design thinking

## Abstract

The current approach to stimulating elderly physical activity mostly uses medical rehabilitation methods or popular forms of outdoor recreation. In the context of an ageing population, there is a growing demand for innovative rehabilitation methods that use information technology. In this article, we present the Urban Health Path as an innovative form of activation for older people using urban therapy, where the architectural elements, such as details, façade features, and urban furniture, inspire movement and attentiveness in the experience of space. The concept is supported by a mobile application that takes into account the specific preferences of older users. Our concept of the physical and cognitive activation of older people was the result of a user-centred design approach and it was tested as a prototype solution. At the same time, the aim of this article is to identify opportunities and limitations for the implementation of this type of solution in other urban spaces. The article presents the process of developing a solution using the Design Thinking method. The process was focused on the needs and preferences of older people. The results of the research project indicate the main important guidelines for implementing the Urban Health Path as a new form of urban facility in the city space.

## 1. Introduction

The ageing of the population and the growing need for the rehabilitation and maintenance of the psychophysical well-being of older citizens call for innovative and engaging forms of physical and cognitive activation. This includes not only taking advantage of the latest developments in physiotherapy, the development of technological and communication devices, or social gerontology, but also adopting a methodology for designing products, services, or spaces that are orientated towards the needs and preferences of the user. Regarding the urban ageing process, urban planning is currently strongly focused on the idea of active ageing [1,2,3,4]. To increase physical activity among older people, the characteristics of urban spaces and especially green spaces are used as areas with significant potential to shape public health. These objectives are being pursued through a number of urban policies and strategies in the domains of age-friendly cities: community and health care, transportation, housing, social participation, outdoor spaces and buildings, respect and social inclusion, civic participation and employment, communication, and information. These policies include the WHO Age-Friendly Cities Framework [5] and the Active Ageing [6] and AGE Platform Europe [7] programmes. In addition, the concepts of universal design and active design are currently being used in the design of age-friendly spaces, promoting, among other things, the development of active mobility and accessible pedestrian zones in cities and buildings [8,9].

Previous research in the field of physical activity for the elderly has focused on the characteristics of recreational areas that promote physical activity and build self-esteem [10,11,12,13], improving social integration in the context of urban green space visitation patterns [14,15]. Past research was also focused on the importance of urban design in implementing the idea of active ageing in the range of building local identity and attaching it to a place [16,17]. The role of the association between barriers in the outdoor environment and perceived quality of life (QoL) in old age was underlined [18,19,20]. Several individual factors were identified as limitations to moving outdoors, e.g., mobility difficulties due to musculoskeletal conditions or fear of going outdoors, including limitations caused during the COVID-19 pandemic [21,22]. An important factor that also affects the quality of life of older people is social capital, social activities that strengthen bonds and significantly improve quality of life [23]. Important research results also indicate the role of social and physical activities, community life facilities and services, social networks, and a clean and pleasant environment as important factors that should be considered in enhancing the social well-being of the elderly in urban renewal processes [24]. The importance of physical activity and its supportive role in quality of life has been presented in a variety of lifestyles [24,25,26], home interventions, and environmental support to increase physical activity [27,28]. Furthermore, the role of technological tools in promoting active ageing has been strongly underlined [29]. Research on assistive eHealth technologies and interfaces for outdoor therapeutic spaces is in its nascent stages and has limited generalisability [30,31,32,33,34]; however, the importance of such tools for enhancing social interaction has been clearly highlighted [29,34,35]. An analysis of the impact of mobile applications on increasing physical activity levels among adults showed that significant effects can be observed when applications are combined with health coaching, personalised text messages, and self-monitoring components [33]. No studies have shown a direct effect of the use of health-promoting apps on the treatment of chronic diseases in the elderly, i.e., dementia [31]. On the other hand, the effect of delaying the development of cardiovascular disease, lowering blood pressure levels, increasing cardiac stroke volume, making the arteries more flexible, and reducing the risk of diabetes and atherosclerosis as a consequence of physical activity has been repeatedly confirmed [29,36]. It has been confirmed that digital tools that mobilise movement can significantly support other treatments and improve quality of life [36,37,38]. On the other hand, there are barriers to using wearable devices among seniors [34,39].

The author’s concept of heritage healing adopted in this work uses the potential of urban architecture, mainly the details of building facades and elements of small architecture. The concept of a therapeutic environment developed so far has been based on the relationship between the user and the elements of the built environment [40,41,42,43], or has been shaped by solutions in line with environmental theories, e.g., biophilic design [44,45]. On the other hand, the perception of architectural elements is designed to stimulate movement by imitating and mapping the shapes of selected elements with movement. A health path was established in a historic city area, highlighting interesting buildings and places important to the local community as exercise points. Therefore, the direct inspiration for the construction of motor and mental tasks is the creative interpretation of architectural elements. The gaze of an art historian, architect, rehabilitator, and physical culture specialist allows a multidimensional elaboration of the potential of individual objects and provides an ascent to alternative forms of reading and use [46]. To date, the physical activity of users in urban space has been developed in the form of tourist routes for active people, supported by a mobility app, popularising historic architecture [47,48]. On the other hand, walking, Nordic walking, and exercise performed in outdoor gyms are predominant among the popular forms of daily physical activity for the elderly [49,50,51,52]. Older people are more likely to participate in activities with small groups of friends, while older people with poorer physical fitness are much more likely to participate in organised activities for safety reasons. It should be noted that individual physical activity is mainly performed in sports or recreational areas: parks, outdoor gyms, bicycle paths, etc. Given the visual system, the musculoskeletal system, as well as cognitive function problems, which progress with age, the combination of targeted mental and physical exercise with walking can contribute to better well-being and health, and slow down degenerative processes [53]. At the same time, movement for seniors also means greater participation in their social life, which unequivocally affects life satisfaction [54,55]. Therefore, the question arises of the potential possibilities of expanding the functionality of urban spaces to improve the functioning of individual organs and systems of the elderly in an integrated way. In addition, the dynamic development of ICT creates additional opportunities to apply digital rehabilitation in the process of improving selected groups of patients [56,57,58,59,60]. Therefore, an important issue is to identify the supporting features of IT tools for the elderly to promote these forms of rehabilitation to users in an attractive and communicative way.

The results of the interdisciplinary Urban Health Path project presented in this article are an extension of research on optimal forms of activation of the elderly following an innovative approach. This article presents the concept of motor and cognitive activation of the elderly (space therapy) and the results of a test of a prototype solution as a result of a user-centred design approach. At the same time, the purpose of this article is to identify opportunities and limitations for the implementation of such solutions in other urban spaces. As part of the implemented project, a senior health path was created, located in the urban space of Gliwice (Upper Silesia, Poland) and linked to a mobile application. The research used a behavioural model of the therapeutic effects of the components of the space [61]. The main scientific goals of the project include the following:Developing criteria for the selection of urban space features (e.g., topography, building elements, or small architecture) with a view to determining their suitability for the rehabilitation of older people;Developing assumptions for the development of technological tools to activate and engage older people in physical activity and cognitive functions;Selecting architectural objects with significant historical and monumental value as elements of user interest in urban architecture.

An important aspect of this project was the introduction of tasks that strengthen the sense of urban identity by attracting residents and visitors interested in the history and architecture of a city, providing opportunities for the group to learn about the values of urban space. In this way, the developed health path not only contributes to physical fitness (improving mobility, motor skills, and coordination of the elderly, reducing the risk of injuries and falls), but also supports cognitive abilities. This may have a supporting role in the prevention of dementia-related diseases. In addition, outdoor exercise increases the body’s resistance to various infections and also reduces stress levels, which, among other things, lead to better sleep quality.

The idea for the project originated from the Active Recovery Foundation in Wrocław, which works for the benefit of oncology patients and promotes a holistic approach to therapy. The project is being carried out by supervisors and students of the Faculty of Automatic Control, Electronics, and Computer Science, the Faculty of Architecture, and the Faculty of Biomedical Engineering of the Silesian University of Technology, supported by external experts in the field of rehabilitation, space therapy, and universal design. The groups of seniors (members of the University of the Third Age and Centrum 3.0—The Gliwice Centre for Nongovernmental Organizations) were also engaged in the testing stage of the project.

## 2. Materials and Methods

At the initial stage of the work, the following substantive assumptions were made:Developing an optimal urban health path programme requires the use of interdisciplinary knowledge from architecture, computer science, physiotherapy, and space therapy;The needs of the elderly as key guidelines in the design process—the use of a person-centred design perspective and enabling the inclusion of various perspectives and user preferences (Design Thinking method);Architecture of historical buildings as an impulse for physical exercise and the intellectual activation of the elderly—public space has morphological, functional, and formal characteristics that can be used as an impulse for physical activity; targeted walks in the historic part of the city combined with activation of the senses, appropriately selected movements, and educational form can contribute to improving health, well-being and, consequently, the quality of life of many user groups:IT tools as an extension of urban functions: it is possible to use mobile tools to provide new urban facilities, introducing innovative forms of engaging the users of public space to experience the city.

The project has identified the following research objectives:(a)In terms of spatial issues,Analysis of the potential of the urban space of Gliwice as an example of a historic city in terms of its use for movement activation; determination of the functionality and features of the urban space and elements of small architecture that may serve movement recreation;Definition of guidelines for the implementation of the urban health path in urban space.(b)In terms of technological issues,Analysis methods of presenting information on smartphone screens adapted to the preferences and possibilities of older people;Exploring the use of wearable devices to monitor the physical activity of app users.(c)Regarding rehabilitation issues,Developing a set of exercises and activation tasks, taking into account spatial conditions, the capabilities of the target group, their potential, and limitations.

The project was carried out in four phases: the objective of the first phase was to develop a prototype solution and conduct tests with a Gliwice seniors group, and in the remaining three phases, the objective was to extend the solution with variant paths, develop a visual identity concept, and extend the application with functionality that allowed the monitoring of user physical activity using mobile devices. The entire process of developing the solution was an iterative process, using both expert knowledge and lessons learnt from the prototype solution testing phases. The methodology adopted is shown in Figure 1. The project adopted the Design Thinking methodology. This is a creative method of developing products, services, and processes using appropriate techniques to generate creative solutions oriented towards the user and their needs. The inter- and multidisciplinary nature of the working team is also characteristic of this method. In addition, it is an iterative process—important elements are the evaluation and verification of the developed solutions in order to achieve the most optimal product or service for the user. A fundamental starting point in the work of the entire team is a deep understanding of how users function and to empathise with their needs. The process involved the following stages [61]: Stage 1: Empathising—research into users’ needsDefinition of hypotheses and their verification during casual interviews: use by seniors, forms of their leisure activities, attitudes toward physical activity, and literature analysis (desk research);Stage 2: Problem definitionGenerating problem statements based on desk research and free interviews to determine the assumptions in the following areas: forms of use of recreational space by seniors, recreational and health needs, specifics of the design of digital tools for seniors;Stage 3: Ideation and exploration of solutionsUrban analysis of the urban space of Gliwice and determination of the route;Defining the functionality of the application and the algorithm for its use, defining the method for monitoring user activity while working with the application, personalising the settings, and defining the statistics collected;Summarising the information collected from the users using the value proposition matrix (needs, problems, and benefits) and generating solutions;Development of application content (exercises and tasks);In situ research on the selected route and key architectural objects;Group brainstorming;Stage 4: PrototypingDevelopment of a prototype mobile application, recording of the content of exercises and tasks, and a preliminary concept for the visual identity of the health pathway;Stage 5: TestingVerification of the developed application and the route of the path with groups of seniors using a test sheet (determination of what works, what should be improved, and collection of questions and suggestions) and use of an application evaluation survey;Stage 6: Implementation planningSummarising the results of testing, grouping errors, and shortcomings that can be solved immediately after testing, and indicating the necessary changes to be implemented in the long term.

### 2.1. Solution Testing Process

As intended, the development of the route of the Urban Health Path was performed in an iterative process with the participation of groups of seniors. The scope of the tests was defined in accordance with the project objectives and with the assumed project work algorithm. All aspects tested were divided into two parts: architectural and spatial aspects, and digital tool functionality. The phases of testing made it possible to determine the general spatial characteristics of the path and refine the solution to the preferences of older people. Furthermore, at each testing phase, the members of the multidisciplinary research team were responsible for defining the main aspects that needed to be improved. Architecture students and teachers were responsible for the spatial solutions and coordination of the testing, the range of exercises with the physiotherapist, and the teacher and students of IT were responsible for application functionality. This led to the development of common conclusions in the form of guidelines that took into account universal design principles. Older women (70 years and older) with good physical fitness participating in the activities of partner NGOs (University of the Third Age in Gliwice and Gliwice for Social Activities Center 3.0) were involved in the subsequent four phases of testing the Health Path. The following issues were evaluated:(a)Phase 1: presentation of the first prototype solution containing the first two routes—Gliwice Market Square and Gliwice City Centre, testing the functionality of the application; the tests included the selection of the locations of exercise points, the concept of movement and breathing exercises and the city game, the length of the routes, the manner of the presentation of content in the application concerning the history of architecture, composition of elevation, detail, and determination of user preferences (*n* = 9 women, January 2022). The test was carried out stationarily on the presentation of the project assumptions and during the use of the prototype version of the app;(b)Phase 2: field testing of the new variant—Gliwice City Centre; testing its route in terms of urban functions (e.g., inclusion of park areas) and selection of exercises, including relaxation exercises (*n* = 7 women, April 2022);(c)Phase 3: desktop testing of the enhanced application functionality; testing in terms of the graphic interface of the application (contrast, background colours, and font size), map functionality, and presentation of exercises for individual points (*n* = 8 women, May 2022);(d)Phase 4: field test of the visual identification system in conjunction with the application; test of the locations of the components in the urban space and their legibility and a field test of the application’s use (*n* = 11 women, July 2022).

### 2.2. Development of the Spatial Layer of the Solution and Exercise Proposal

The paths were developed on the basis of urban planning analyses and spatial valorisation of the city centre. First, well-known architectural objects and city squares were selected as potential exercise points (Figure 2). The selection of locations was then verified during research walks: first by the project team, and then with the participation of senior citizen groups. In the second stage of the investigation (defining the challenge), two variant routes were selected (one shorter, Gliwice Market Square, with a length of 700 m, and one longer, Gliwice City Centre, with a length of 2200 m, Figure 3). An important criterion for the analysis of the routes was the safety and collision-free character of the pedestrian routes and their accessibility. In the concept of the Urban Health Path, movement and relaxation exercises integrated with architectural objects were developed (Table 1) and cognitive aspects were included in the form of an urban game containing trivia and quizzes on selected objects in Gliwice. The components of the health path in urban space were the urban infrastructure and topography of the area of the centre and city centre of Gliwice (pedestrian routes and city squares), recreational public spaces (parks and squares), elements and details of architectural objects (window lintels, portals, pilasters, and texture of facade materials), and elements of small architecture (benches and landings). The results obtained after the first phase of testing led to the development of the route of the longest route, Gliwice Śródmieście, with a length of 3500 m (Figure 3). On each of the routes, the locations of exercise points were selected in the most characteristic places and objects in the space of Gliwice, which were also landmarks of the route or spatial dominants (Figure 2). They were identified as places to perform selected breathing and movement exercises. At the same time, the points served to stop the user on the route of the path and to experience, by means of various senses, the elements of the architecture in a conscious manner.

### 2.3. Testing of Spatial and IT Solutions

The path components analysed in the individual test phases, in accordance with the adopted iterative process, resulted from the adopted objectives and the results of the subsequent test phases. In the first phase, the general features of the solution (spatial and IT elements) were tested. Meanwhile, the indicators in the subsequent phases resulted from the indications of the seniors who participated in the tests. They were segregated into two groups: elements that could be directly implemented in the project and elements that would guide future project development. Although the route was defined along accessible and safe pedestrian routes, the first testing phase showed that the route needed to be modified: the test participants emphasised that they found it problematic to perform the exercises in public spaces (e.g., Market Square). This was related to the feeling of being uncomfortable and being watched in places where this type of exercise had not been done before. During the tests, the following opinions were formulated: “I cannot imagine exercising in the Market Square”, “exercises should be performed in more intimate places (parks, square)”, and “urban space for exercise can be, but not necessarily the strict city centre, e.g., Chopin Park”. During the second phase, the senior women indicated that they were interested in performing the exercises together with the group and that the route path should be designed as a walking route (active walking). They were very keen to perform the exercises in spaces with less pedestrian traffic (Slaughterhouse Square and Bridge of Lovers, Figure 4) and breathing exercises (e.g., in Chopina Park). In general, the level of difficulty of the exercises was described as easy, while it was emphasised that the wording of the exercises had too much medical vocabulary and was not always understandable.

The analysis of the issue in terms of the IT tool began with an examination of the needs and preferences of older people, using literature analyses and free interviews, setting hypotheses in terms of the use of smartphones by seniors in daily life, the importance of physical activity in daily life, the use of the gamification element among seniors, the nature of the seniors’ group activities, and the conditions for feeling safe. Additional aspects that were analysed by the project team were the results of research on how to motivate older people to move, the design of digital tools for seniors, the accessibility principles of the apps (e.g., font size and typeface, forms of assistance, and learning to use the app), and easy and intuitive use. The testing phases of the app, listed in Section 2.1, allowed the app’s functionality to be developed in stages and optimally adapted to the needs of seniors: user interface requirements, feedback, forms of user interaction, and cognitive difficulties in pre-presenting content within the app were defined (Figure 5). Furthermore, methods were developed to monitor the behaviour of older adults based on actions performed within the app, but also using a smartwatch [62]. The data obtained were stored in a database. To obtain relevant data, it was assumed that the use of the smartwatch to monitor selected sensory data was tested.

The testing involved people who use smartphones daily, so their observations related to their previous experience with apps. Taking into account the requirements of senior apps (reducing the number of stimuli and the amount of information flowing from the app, providing only the necessary functionality), it was decided to simplify the use of the app in terms of personalising the interface to the user’s needs (changing font size and colour theme). Instead of changing the colours of individual elements, seniors only had to make a single selection of the colour theme describing all the colours of the application’s interface (Figure 6). In addition, changing the font size and colour theme did not require additional approval, thus reducing the number of actions that users needed to perform. At this stage, information about changes to interface settings was stored in a database.

Work on the implementation of the sensory data measurement system began with the creation of a “wearable” application for the watch under test (Samsung Galaxy Watch 4 Classic), acquiring data from the built-in sensors. Owing to difficulties with the use of Samsung Health’s proprietary SDK, manual data extraction was performed by creating a watch-based exercise linked to the health tracking, monitoring the following selected parameters:Number of calories burnt;Heart rate;Distance covered;Number of steps.

Then, an information channel was established to transmit the measurement results to the mobile application for further processing and transmission to the database. As previously agreed, a low-level communication system was developed using the Bluetooth standard for correct recipient identification and data transfer. At this stage of the work, the application could be launched in two ways, automatically and manually. Automatic launch occurred on any watch paired with a phone using Bluetooth technology at the beginning of a track. While the application was running, the user could use a single button to start and stop the transmission of information to the mobile application.

### 2.4. Assumptions Made in the Physical Activity Method for Older People

In line with previous assumptions, the specifics of the age group were taken into account during the stage of developing the physical exercises, including the fitness, capabilities, and limitations of the elderly. Possible dysfunctions of the musculoskeletal system, as well as the safety of the user group, were also taken into account in the development of the kinesitherapy parameters. The aim of the prepared set of exercises was to stimulate various systems, especially the respiratory system and proprioception (e.g., balance and coordination). Improving the efficiency of the respiratory system leads to better oxygenation of, among other things, the muscles and the brain, which is crucial in older people. Improved blood flow in blood vessels, in turn, supports heart function. Due to the broad target group and the varying levels of fitness, individual exercises were differentiated in terms of execution difficulty. The emphasis of training was on mixed cardio-respiratory training. The selected exercises that were performed in the project were prepared on the basis of the most frequently reported back, lower, and upper limb pains of patients in the rehabilitation clinic. They served to improve health and injury prevention. At the same time, the exercises were inspired by the shapes of architectural elements and reflected the kinaesthetics of the spaces encountered, making them innovative and unprecedented in the hitherto implemented movement therapies. The exercises were neither difficult nor very complicated and could be performed without special equipment or prior preparation. They have a positive effect on prolonging the period of independence and are one factor in the prevention of falls. In this way, seniors can enjoy a longer life and avoid one of the most common causes of fractures. It is important to maintain or improve muscle function in the upper and lower extremities, as this is associated with improved performance in activities such as walking quickly, climbing stairs, and standing with balance.

## 3. Results

The implementation of research in the field of analysing the ways in which urban space can be used for its health-promoting potential and modern IT technologies can be used for piloting and monitoring project participants has provided knowledge on the implementation of innovative solutions to improve the psychophysical condition of older people. The innovative solution in the form of the Urban Health Path is an example of the possibility of developing digital urban services for recreation and active leisure for older people while, at the same time, activating urban spaces (e.g., by creating alternative walking paths). The physical and cognitive activation of older people was carried out through developed exercise sets using details and elements of urban architecture as a stimulus for exercise.

The iterative process adopted resulted in a solution that took into account the needs and preferences of seniors in terms of the forms of activity, the specificity of physical exercises, and the adaptation of the mobile app to the perceptual capabilities of seniors. The scope of the tests was defined in accordance with the project’s objectives and is described in Section 2.1. After the first test phase, the exercises in public spaces were modified to be less noticeable. After the second phase, a new route was also proposed, including squares and city parks. The structure of the application was modified by placing buttons in more intuitive positions on the screen. The third phase supplemented the app with a summary of the route taken. The fourth phase resulted in the development of supporting materials (brochures and an identification system).

The selection of building elements and architectural details as exercise impulses were guided by their expressiveness and the shape that could be mimicked during the exercise. Examples of exercise sets are included in Table 2. In the initial phase of the project, close cooperation was established with rehabilitation experts, who determined a programme of movement and relaxation exercises integrated with architectural elements.

### 3.1. Urban Path Design Guidelines for Spatial Solutions

The implementation of this project, including the testing phases with seniors, allowed the development of guidelines to replicate the solution in the spaces of other cities, which is one of the most important scientific results of this project. Guidelines for spatial solutions can be divided into three groups of issues:

A1. Diversity of types of exercise and intensity as one of the basic factors determining the flexibility of the solution.

Route variants in terms of distance;Close proximity of individual exercise points, free choice of the number of exercise points on the route;Zoning of the route in terms of exercise intensity: exercise sequences should be more intensive in urban green areas and less demanding and visible in public spaces (pedestrian traffic areas);Selection of interesting architectural details—selection of breathing exercises (relaxation) as an element of mindfulness enhancement.

A2. Clear delineation of exercise areas.

Use of characteristic details, sculptures, etc.;Integrating the names of the exercise points with characteristic and recognisable elements of the space;Introduction of permanent elements of the spatial identification of the path to mark the route and exercise points in space: introduction of small architectural elements; their form and colouring should be integral to the project’s logo.

A3. Accessibility of urban space in terms of pedestrian routes.

Even and slip-resistant surfaces for pavements and squares;Reduction in collisions of pedestrian routes with traffic by means of tactile elements, colours, or changes in paving;Selection of conflict-free exercise areas (e.g., outside the pedestrian flow routes)—recesses, pavement widenings, and sections of city squares outside the main pedestrian flow routes;Use of existing rest areas (e.g., benches, support points e.g., walls).

Furthermore, the guidelines developed take into account urban architectural features that motivate movement and active exploration of space, directing the user to be attentive in observing their surroundings.

B1. Information on the morphology of an object contained within its description: form, composition, and detail, and learning and understanding basic architectural concepts, such as mental training.

B2. Elements of buildings and architectural details of buildings that can be integrated with movement (e.g., arcades, rhythm of columns, rhythm of windows, pillars, pilasters, turrets, bays) and integration of movement with the shape and rhythm of architectural elements of buildings.

B3. The complexity of historic buildings and features of buildings to exercise concentration and imagination, e.g., recreating memorised details in the imagination, imaginatively reducing and filling in window shapes, and following the complex line of cornices with the eye.

B4. Elements of urban spaces and details of buildings stimulating particular senses, such as touch (e.g., textures of façade materials and surfaces of pedestrian walkways), hearing (e.g., the sound of water (fountains) and the sound of the clock on the town hall tower), sight (e.g., the colours of façade materials), and smell (e.g., smells in the vicinity of restaurants and bakeries).

### 3.2. Guidelines for Designing an Urban Pathway for IT Solutions

Guidelines for IT solutions for seniors should be consistent with the WCAG guidelines in terms of perceivability, functionality, comprehensibility, and compliance. The guidelines can be divided into:

(1) Requirements for the user interface in the application—in particular, the following:Buttons allowing unobstructed operation by seniors and large spacing between buttons;Large and legible font (avoiding bold, italics, and animated text) and contrast with the background;Contrasting colours;Limited animations on the interface (to reduce distractions);Simplified input method (reducing the need to type text);Interaction with the application at the level of easily identifiable interface elements (e.g., buttons);Introduction of interface customisation possibilities.

(2) Application performance requirements, in particular:Slower, subdued app mechanics;Limited number of stimuli and amount of information flowing from the application;Providing only the necessary functionalities;Limited information on application errors;Frequent positive feedback;Use of “soft delete”—any significant change can be undone;Introduction of automated operations (reduction in the number of decisions and actions performed by seniors).

(3) Requirements to make the application environment and its rules easy to understand, in particular:Simple, easily understandable mechanics and rules of the application;The iconography of the application should refer to the daily life of seniors;The behaviour of the application should be compatible with the tools used by seniors;Provision of help describing how to use the application.

## 4. Discussion

The implementation of the project demonstrated the potential for the innovative use of architecture as a space for physical exercise and engaging the perception of older people. The concept of a healing environment, which makes use of selected architectural features of therapeutic importance (e.g., the importance of colour, lighting, and elements of nature), is mostly associated with intentional design activities that shape the optimal features of the physical environment and the environment–occupant–health (E-O-H) model [43]. Furthermore, it is mostly directed at the passive effects of environmental characteristics on human health. Although research in this area does not explicitly point to a relationship between environmental factors and the health process, it confirms that this process is the product of emotional state, cognitive processes, and physical fitness [24]. In this sense, the solutions used in the project represent an innovative approach and aim to improve health through the activation of older people. Its practical importance should be considered in relation to the following aspects of improving and maintaining health:Proposition of engaging physical exercises involving kinaesthetic elements (e.g., active observation and imitation of object features) and sensory elements (e.g., feeling the texture of façade materials with the hand), which are important for improving older people’s overall motor coordination and triggering the activation of the conscious perception of the environment, which is meaningful for their short-term memory and as an exercise in focussing attention;Selection of relaxing exercises involving perception and attentiveness in observing the environment (e.g., recreating in memory the shape of the head of a portal column)—a significant element in preventing dementia and engaging the attention of seniors and their short-term memory;Use of the mobile app as a motivational tool, presenting feedback in-app (e.g., length of route covered, determination of pulse changes, and duration of physical activity)—this serves as motivation for regular physical activity—and obtaining data on basic health parameters to evaluate overall health; moreover, this could be the element of improving digital competences of seniors;Strengthening the sense of local identity, attempting to improve interest in the history and architecture of the city, and learning together as a group impulse—social activities in a group of seniors strengthen ties significantly and potentially improve quality of life. In addition, the project could be used as a tool for the Centre for Local Activities in special forms of activities for the group of seniors.

These solutions add a new dimension to the impact of space on users, introducing the concept of urban therapy with its many potential health-promoting and activating factors. According to the assumption, the presented solution is the result of combining knowledge from several areas. The selection of architectural objects and the route required knowledge of architectural history, universal design, rehabilitation, and environmental psychology. Buildings that are well known to the local community, with distinctive architectural façade elements, and that are landmarks along the route were chosen as training points. This was also important to strengthen the sense of local identity through the introduction of new activities (physical exercise) combined with a conscious perception of urban architecture. Of particular interest to senior test takers was the developed urban game, which encouraged walking and learning about the history of the city. It should be noted that, at individual points along the route, the seniors paid attention to the architectural values of the selected buildings, their details, and the atmosphere of the place. A place of particular interest and discussion was the ruins of the old theatre on Friendship Avenue, whose history was less well known to residents. At the same time, the selection of points was dictated by local conditions (e.g., the way the selected spaces were used) and features allowing for safe and comfortable exercise (e.g., avoiding locating exercise points in areas of heavy pedestrian traffic and avoiding routes in places of collision with traffic). Photographs of objects or details were, at the same time, the icon of the exercise point in the app, to make it easier to calibrate the points in the city’s structure. An important cognitive aspect in the presentation of the exercises was the explanation of concepts such as rustication, pilaster, and plinth as part of the basic information about the object, and the subsequent use of this architectural element in the exercises.

An important outcome of the project was that the test seniors confirmed the attractiveness of the tested solution, i.e., the linking of the path elements with the mobile application. Other studies have also shown that it is possible to increase motivation for physical activity among seniors [30,63,64]. The app provided instructions for performing exercises (videos and text) and mobilised movement through the function of individual features. These include the possibility to obtain feedback on the result of exercise (distance covered), the possibility to share this information with friends, and the possibility to obtain basic sensory data (number of calories burnt, heart rate, distance covered, and number of steps).

A study by Kabisch, Puffel et al. [12] showed that activity in urban green spaces has a beneficial effect on the psychophysical state of older people, especially on their cardiovascular systems. This fact was used in the project, especially because an important result was the clear preference to exercise more intensively in urban green areas. This influenced the inclusion in the guidelines (Section 4.1) of a differentiation of the intensity level of exercise in relation to the function of urban areas, so that more intensive exercise was planned, for example, in urban parks.

Functionality testing of the mobile app with groups of seniors also found that testers expected to be able to operate the app using gestures instead of buttons (contrary to the literature on the subject). However, this confirmed the principle that senior citizens are not a homogeneous group with the same preferences; they were used to operating mobiles with gestures due to their experience with phones. The small size of the test group and the specificity of the group (participants of the University of the Third Age, active people focused on new challenges) do not allow the results to be generalised to a larger population of senior citizens. However, to test this observation, the possibility of changing font size using a slider was added to the design of the application.

### 4.1. Limitations

In the tests carried out, due to the specific nature of the participants in the activities of the partner NGOs (their activities involve a great majority of women), only older women participated, which is a limitation in obtaining an objective test result. The project did not test users’ perception of the visibility of individual architectural elements, but the selection of individual elements was guided by their large scale and expressiveness (e.g., contrasting colours of the detail against the colours of the facade material).

The mobile application itself did not allow for a detailed analysis of the impact of the health path on the physical condition of the elderly, and thus made it challenging to identify potential problems with the path; for example, exercise can be too strenuous for the elderly. Due to the specific nature of the participants in the activities of the partner NGOs (their activities include an overwhelming majority of women), only older women took part in the research, which is a limitation in obtaining an objective research result.

This project did not examine users’ perceptions of the visibility of individual architectural elements; instead, the selection of individual elements was guided by their large scale and expressiveness (e.g., contrasting colours of the detail with the colours of the façade material).

### 4.2. Future Work

Testing the effectiveness of the spatial and IT guidelines in the example of other cities and testing the solution with a representative group of seniors are priorities for future research One of the directions of future research is to improve data collection for a better assessment of the behaviour of seniors (that is, developing methods to integrate sensory data from wearable devices with the behaviour of seniors during health pathway testing). We could also introduce surveys to mobile applications to profile users in more detail (e.g., health problems and number of close friends). Furthermore, future work could define the spatial factors driving physical activity and interest in local spatial values in culturally diverse local communities.

An important aspect of further research is the possibility of personalising the exercise programme and the application interface to the needs and preferences of seniors. An outline research programme could include the testing of new exercise sets, grouped according to difficulty and typical locomotor disorders. In addition, given the results of the research, the level of intensity of the exercises should be chosen in terms of the visibility of the exercises to passers-by to increase users’ comfort.

From the point of view of the importance of the project to strengthen local ties and the concept of becoming a resident, a possible research direction is the analysis of cultural and spatial factors that determine the acceptance of physical activity by older people in public spaces.

## 5. Conclusions

In this article, we present a concept for the motor and cognitive activation of older people in the context of walks in urban spaces. The solution developed in the form of an Urban Health Path is based on the potential of Gliwice’s historic buildings and the authors’ method of urban therapy. Analyses of the architectural features of buildings and street furniture, as well as the testing phase with senior citizens, have led to the development of guidelines for the application of this solution to other cities in terms of spatial and IT conditions. The Urban Health Path is mapped in urban spaces and integrated into mobile applications. First, it aimed to increase the attractiveness and accessibility of the solution. It had the functionality of presenting the content of exercise and architectural history and of monitoring physical activity in the application. The solution is important for increasing physical activity among the elderly and strengthening local identity by spending leisure time together in attractive urban spaces. The project demonstrates the new potential of urban spaces and the tests carried out show that it could also increase the attractiveness of physical activity for seniors. The project has a number of possible directions for further research, primarily into the effectiveness of the method in increasing physical activity, the personalisation of the tool, and the cultural determinants of physical activity in urban public spaces.

## Figures and Tables

**Figure 1 ijerph-20-06081-f001:**
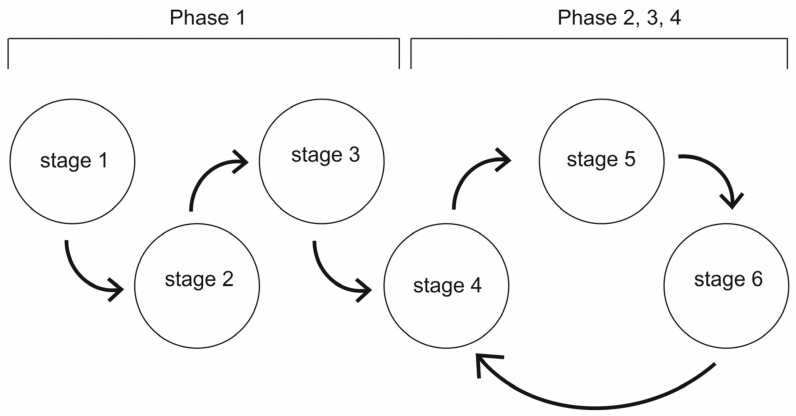
Diagram of the project implementation process.

**Figure 2 ijerph-20-06081-f002:**
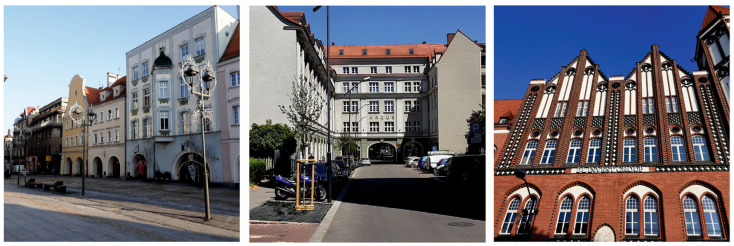
Examples of urban space objects in Gliwice used in the Urban Health Path project; from left: northern frontage of Market Square; buildings along Gruszczyńskiego Street; and fragment of an elevation of the former post office building. Photo: A. Szewczenko.

**Figure 3 ijerph-20-06081-f003:**
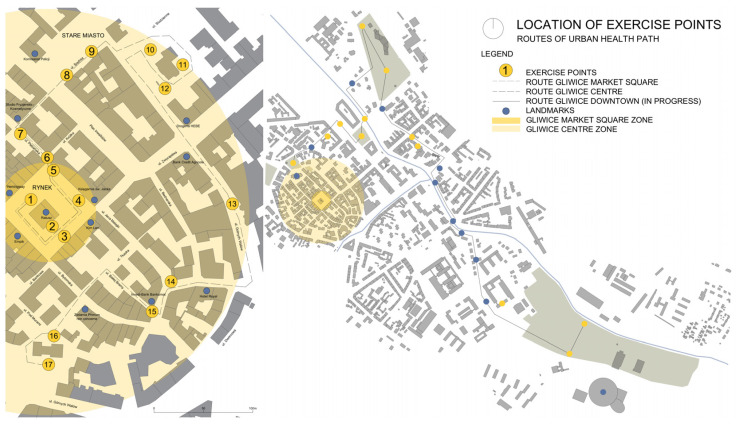
Routes of the health paths. On the left: Gliwice Market Square and Gliwice City Centre routes, on the right: Gliwice City Centre route. Prepared by: M. Sanigórska, D. Bal, and K. Elsner.

**Figure 4 ijerph-20-06081-f004:**
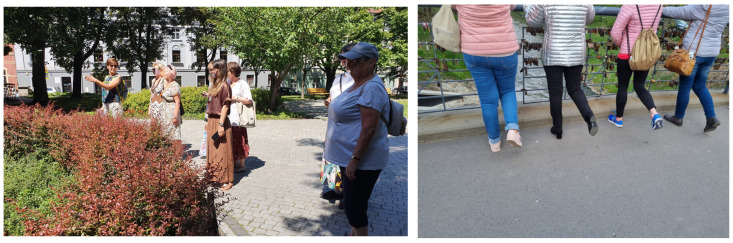
Photographs from the Urban Health Path field test run, April and July 2022. Photo by: I. Chuchnowska.

**Figure 5 ijerph-20-06081-f005:**
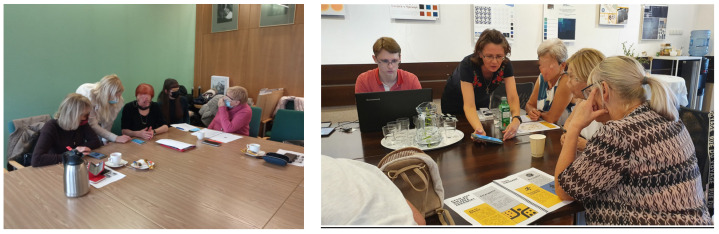
Application testing process, January 2022. Photo by: A. Szewczenko.

**Figure 6 ijerph-20-06081-f006:**
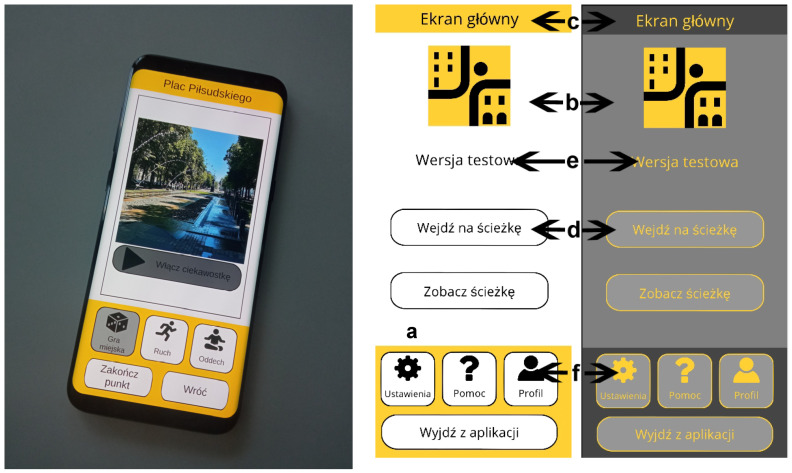
Appearance of the application’s interface. Elements of the interface that can be modified by the user: a. font size, b. basic background colour, c. complementary background colour, d. button colour, e. font colour, f. icon colour. Photo by: **left**, A. Szewczenko; **right**, M. Sutor, J. Włodarz.

**Table 1 ijerph-20-06081-t001:** The most important indications for modifying the prototype in the various test phases.

Testing Phase	Test Participants’ Indications	
	Spatial Solutions and Range of Exercises	Application Functionalities
Phase 1: Prototype test *n* = 9	Determining the location of path points in the area Restrictive exercise on the Market Square—location of exercise points in green areas or recreation areas Attractive form of active leisure Attractiveness of the city game—expanding information on the history of the city	Possibility of obtaining information on the number of calories burnt Change from map with paths to active map Application needs to be improved to be more intuitive Unreadable and inactive map Choice of voice or text messages
Phase 2: Field test *n* = 7	Recreational areas and the city park as comfortable places to exercise In public spaces, comfortable places to exercise with little visibility Favourable exercise locations that do not interfere with pedestrian traffic Supplement the application with a paper map It is important to link the exercise point to the history of the place in order to establish a relationship with the place	Audio files difficult to hear amidst the urban bustle of the Market Square
Phase 3: Stationary test *n* = 8	None	Volume control most convenient via phone buttons Completion of location recording for real changes in location in the field Supplementation of route information (length and waypoints)
Phase 4: Field test *n* = 11	Enlargement of the signs marking the exercise points, which were placed too low Helpful brochures with information on the route and exercises	Positive results of the smartwatch-linked application test for data collection and gathering

**Table 2 ijerph-20-06081-t002:** Example exercises integrated with selected architectural elements.

Selected Architectural Element	Brief Description of a Sample Exercise	Photo-Documentation of the Exercise Point
Arcade arches in a historic building	Imitating the arches of the arcade with circular hand movements while walking along the building	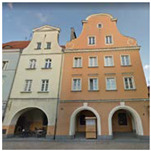
Triangular window lintels	Mimicking the shape of the lintel through head movement with cervical spine activation	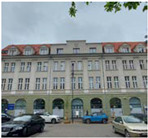
Rhythm of the pilasters on the façade	Rhythmic lifting of the knees while walking along the building according to the rhythm of the pilasters on the façade	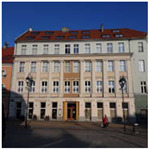
Rustication on the facade mantelpiece	Bending the upper part of the trunk in rhythm according to the stretches marked by the bony line	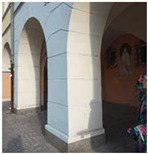
Fountain base	March backwards along the fon-tanna pedestal while maintaining a running track	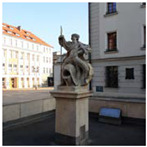

## Data Availability

Not applicable.

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
