# Peer review of "Urban Therapy—Urban Health Path as an Innovative Urban Function to Strengthen the Psycho-Physical Condition of the Elderly"

_ijerph, 2023, doi:10.3390/ijerph20126081_

Round 1
Reviewer 1 Report
Dear authors,
Thank you for this inspiring research and interesting manuscript. The context, methodology, data analysis, results and conclusions are sound. The urban path design guidelines are very compelling and you mention that it is the result of the testing phase. It would be interesting though to mention how the multidiciplinarity of the team contributed to shaping and fine-tuning the final guidelines.
Under limitations 4.1 two paragraphs are repeated twice, please revise.
One comments regarding figure n.6, "a. font size" cannot be located on the figure.
Author Response
First of all thank you for your work in revising our paper. We are grateful for your valuable feedback. We addressed the suggestions and made the changes within the manuscript.
We have, accordingly, mentioned how the multidisciplinarity of the team contributed to shaping and fine-tuning the final guidelines to emphasize this point:
Moreover, at every testing phase the members of multidisciplinary research team were responsible for defining the main aspects needed to be improved. Students and teachers of architecture were responsible for the spatial solutions and for the coordination of testing the range of exercises with physiotherapist, and teacher and students of IT were responsble for application functionalities. This led to the development of common conclusions in the form of guidelines taking into account universal design principles. The repeated paragraph in the guidelines (point 4.1.) was removed. Also, the missing designation a. font size on the Figure 6 was completed, placed by button “Settings” (Ustawienia), which has the functionality of changing the font size.
Reviewer 2 Report
this manuscript presents a concept for motor and cognitive activation of older people in the framework of walks in urban space.
The research content has significant practical significance. Especially for research on the elderly, there is still a lack of quantitative conclusions at this stage. This manuscript addresses these issues to some extent.
However, there are also some areas that need to be revised during the writing of the paper.
1) When reviewing existing research, one should not cite too many literature when describing a particular viewpoint and should clarify the similarities and differences between these pieces of literature. This issue has appeared multiple times in the introduction section of the paper.
2) The paper did not provide a good description of the practical significance of the conclusions found in this manuscript during the discussion section. The current chapter mainly describes the findings of the paper without explaining its purpose. It is recommended that the author add more content to this section.

Author Response
First of all thank you for your work in revising our paper. We are grateful for your valuable feedback. We addressed the suggestions and made the changes within the manuscript. First of all, we have modified the literature review according to your suggestions. Existing research is presented more precisely, indicating in more detail the issues related to the theme of the work (i.e. the role of building the social capital among the elderly, the supporting role of the technological tools in promoting active ageing).
We added also the necessary additions in chapter 4. Discussion. It means that every four aspect of the presented significance got the comment indicating the potential influence on improving health:
- proposition of engaging physical exercises involving kinaesthetic elements (e.g. active observation and imitation of object features) and sensory elements (e.g. feeling the texture of façade materials with the hand) - important for improving older people's overall motor coordination and triggering the activation of conscious perception of the environment, meaningful for their short-term memory and as an exercise in focussing attention,
- selection of relaxing exercises involving perception, attentiveness in observing the environment (e.g. recreating in memory the shape of the head of a portal column) - a significant element in preventing dementia, engaging the attention of seniors and their short-term memory,• use of the mobile app as a motivational tool, presenting feedback in-app (e.g., length of route covered, determination of pulse changes, duration of physical activity) - motivation for regular physical activity, obtaining data on basic health parameters as a check on overall health; moreover, this could be the element of improving digital competences of seniors,
- strengthening the sense of local identity, attempt of interest in the history and architecture of the city, learning together as a group impulse - social activities in a group of seniors strengthen ties significantly and potentially improve the quality of life. Moreover, the project could be used as a tool for Center for Local Activities in special forms of activities for group of seniors.

Reviewer 3 Report
Authors have done lots of work to implement a project to provide urban therapy for the elderly, which is quite an interesting idea. I think this manuscript is ready to be published after making few minor revisions as follows.
What is Design Thinking methodology, and why authors adopted this methodology, please explain.
How authors determined the indicators in Table 1, and what are the results of these indicators, please show them in the part of results.
Minor editing of English language required.
Author Response
Thank you for taking the time to assess our manuscript. We are also grateful for your insightful comments and feedback. We have been able to incorporate changes that are highlighted within the manuscript. The concept and methodology of Design Thinking was explained, as suggested (Chapter 2. Materials and Methods. ("It is a creative method of developing products, services and processes using appropriate techniques to generate creative solutions oriented towards the user and their needs. The inter- and multidisciplinary nature of the working team is also characteristic of this method. In addition, it is an iterative process - important elements of it are the evaluation and verification of the developed solutions in order to achieve the most optimal product or service for the user. A fundamental starting point in the work of the entire team is a deep understanding of how users function and empathising with their needs.").
In chapter 3. Results we explained also determinig the indicators included in Table 1, and in presented results we pointed the main modifications and results after every test phase:
The scope of the tests has been defined in accordance with the project objectives and is described in section 2.1. All tested aspects were divided into two parts: architectural and spatial aspects, and digital tool functionality. We presented the results of every testing phase as follows:
The iterative process adopted resulted in a solution that took into account the needs and preferences of seniors in terms of forms of activity, the specificity of physical exercise and the adaptation of the mobile app to the perceptual capabilities of seniors. After the first testing phase, exercises in public spaces were modified to be less noticeable. After the second phase, a new route was additionally proposed, including squares and city parks. The structure of the application was modified by placing buttons in more intuitive positions on the screen. The third phase supplemented the app with a summary of the route taken. The fourth phase resulted in the development of supporting materials (brochures, identification system).
We also made linguistic corrections to the text.